

# Humidity changes and possible forcing mechanisms over
# the last millennium in arid Central Asia
Shengnan Feng[1], Xingqi Liu[1*], Feng Shi[2,3,4], Xin Mao[1,5], Yun Li[6], Jiaping Wang[1]
[1] College of Resource, Environment and Tourism, Capital Normal University, Beijing 100048, China
[2] Key Laboratory of Cenozoic Geology and Environment, Institute of Geology and Geophysics, Chinese
Academy of Sciences, Beijing 100029, China
[3] Georges Lemaître Centre for Earth and Climate Research, Earth and Life Institute, Université
Catholique de Louvain, Louvain-la-Neuve 1348, Belgium
[4] CAS Center for Excellence in Life and Paleoenvironment, Beijing 100044, China
[5] Institute of Hydrogeology and Environmental Geology, Chinese Academy of Geological Sciences,
Shijiazhuang 050061, China
[6] Qinghai Institute of Salt Lakes, Chinese Academy of Sciences, Xining 810008, China
*Correspondence to*: Xingqi Liu (xqliu@cnu.edu.cn)
**Abstract**
Hydroclimate changes have exerted a significant influence on the historical trajectory of ancient
civilizations in arid Central Asia where the central routes of the Silk Road have been hosted. However,
the climate changes at different time scales and their possible forcing mechanisms over the last
millennium remain unclear due to low-resolution records. Here, we provide a continuous high-resolution
humidity history in arid Central Asia over the past millennium based on the ~1.8-year high-resolution
multiproxy records with good chronological control from Lake Dalongchi in the central Tianshan
Mountains. Generally, the climate was dry during the Medieval Warm Period (MWP) and Current Warm
Period (CWP), and wet during the Little Ice Age (LIA), which could be attributed to the influence of the
North Atlantic Oscillation (NAO) and the Atlantic Multidecadal Oscillation (AMO). Furthermore, we
find that the humidity oscillation was dramatic and unstable at multidecadal to century-scale, especially
within the LIA. The continuous wavelet analysis and wavelet coherence show that the humidity
oscillation is modulated by the Gleissberg cycle at the century-scale and by the quasi-regular period of
El Niño-Southern Oscillation (ENSO) at the multidecadal scale. Our findings suggest that the effect of
the solar cycle and the quasi-regular period of ENSO should be seriously evaluated for hydroclimate
predictions and climate simulations in arid Central Asia in the future.
**1 Introduction**



Arid Central Asia (ACA), far away from the ocean, is not only one of the driest and largest inland
regions worldwide, but also the birthplace of the far-reaching ancient civilizations that spread along the
Silk Road (Li et al., 2016; Narisma et al., 2007). Scarce precipitation, intense evaporation and fragile
ecosystems render this region sensitive to abrupt changes in effective humidity. Proxy records from ACA
are valuable for understanding the driving factors and processes of underlying hydroclimate evolution in
the inland region, which provides useful reference for human adaptation to hydroclimate changes at
present and into future (Li et al., 2016), yet very few high-quality records exist over the past millennium.
Several records from ACA show similar climate patterns over the past millennium, i.e., a relatively dry
Medieval Warm Period (MWP) and Current Warm Period (CWP), and a wet Little Ice Age (LIA) (Chen
et al., 2015; Chen et al., 2010; Chen et al., 2019a; Chen et al., 2019b). However, the climate changes at
short-time scale are unclear. Additionally, some debates occur regarding the driving mechanism of
natural hydroclimate evolution over the last millennium in ACA. Many studies emphasize that internal
climate variability (i.e., atmosphere-sea interaction) is supposed to be more marked than external factors
(e.g., solar forcing) in influencing hydroclimate changes (Chen et al., 2019a; Chen et al., 2006; Lan et
al., 2019). In contrast, several paleoclimate records prefer to highlight the possible solar forcing to the
humidity oscillation over the last millennium (Song et al., 2015; Zhao et al., 2009; He et al., 2013; Ling
et al., 2018). These debates are largely attributed to low-resolution records or inevitable chronological
uncertainties due to the old carbon effect (Chen et al., 2010).
Here, we present a continuous ~1.8-year high-resolution humidity reconstruction with good
chronological control from Lake Dalongchi in the central Tianshan Mountains of ACA over the past
millennium (1180-2018 CE). Low- and high-frequency signals recovered from our reconstruction offer
the potential to detect climate fluctuations at decadal to centennial scales, as well as long-term changes.
Then, we explore the contribution of internal climate variability and solar activity to humidity oscillations
at different timescales over the past millennium.

**2 Study site**

Lake Dalongchi (83°16'48"~83°18'15" E, 42°26'31"~42°26'58" N, 2400 m above sea level) located
on the south slope of the central Tianshan Mountains, is an ideal location for investigating hydroclimate
changes in ACA (Fig. 1a), as the lake sits in the core area of the 'westerlies dominated climatic regime
(WDCR) ' (Chen et al., 2019a). It is an alpine freshwater lake formed by glacial moraine damming and



has a mean pH of 8.03 and a salinity of 0.31 g L$^{-1}$. The lake is mainly fed by the surrounding river and
precipitation within the catchment and glacial meltwater from the surrounding mountains at high
elevations. The lake water flows out into Lake Xiaolongchi through the underground river. The lake
water from Xiaolongchi flows westwards into the Kuche River basin (Fig. 1b). Measured in July 2018,
the lake covers a surface area of ~1.4 km$^2$ and has a catchment area of ~131 km$^2$. The maximum water
depth of Lake Dalongchi reaches 7.4 m in the western lake and its depth largely decreases from west to
east (Fig. 1c). Alpine coniferous forests dominated by Picea primarily thrive on the south and west slopes
of the surrounding mountains, while the north slope is dominated by shrubs. According to the records
from 1958 to 2000 at Bayanbuluk Station which is located ~100 km to the northeast of Lake Dalongchi,
the mean annual temperature is -4.6 °C with a July average of 10.9 °C and a January average of -26.4 °C,
and the mean annual precipitation and evaporation are 270 mm and 3200 mm, respectively. The mean
annual precipitation is far less than the evaporation capacity. Moreover, most of the precipitation occurs
as convective rainfall from June-August (Lan et al., 2018).
**3 Materials and methods**
**3.1 Sampling**

In July 2018 and August 2019, we retrieved several parallel long sediment cores from Lake

Dalongchi at a maximum water depth of 7.4 m using the UWITEC platform manufactured in Austria,
and several short cores using a piston gravity corer (Fig. 1c). To determine overlaps and ensure the
continuity of the cores, magnetic susceptibility (MS) nondestructive scanning of all long and short
sediment cores was used to obtain a 6.95m-long composite core (DLC1819) (Fig. 2). The composite core
is mainly composed of clayey silt and can be divided into two lithology units from bottom to top. Unit A
(695-430 cm) is mainly composed of dark or light brown clayey silt with clear lamination. Unit B (430-
0 cm) has a dramatic lithological variation characterized by brown or greyish-green clayey silt and no
obvious lamination (Fig. 2).
**3.2 Laboratory analysis**

DLC1819 core was subsampled at 1-cm intervals. All subsamples were stored in a freezer in our

laboratory at a constant temperature of 4 °C and used for MS, grain size, total organic carbon (TOC), and
total nitrogen (TN) analyses. Samples for MS were dried at below 40 °C in a constant temperature air-
blast drying oven and ground and packed into standard plastic boxes with a capacity of 2*2*2 cm$^3$. A
Bartingon MS2 susceptibility meter was used to measure the sample mass susceptibility. Samples used
for grain size measurement were pretreated with 10-20 mL of 10% $H_2O_2$ to remove organic matter and
then with 10 mL of 10% HCl to remove carbonates; next, samples were rinsed with deionized water and
finally treated with 10 mL 0.05 M $(NaPO_3)_6$ on an ultrasonic vibrator for 10 min to promote dispersion.
Grain size distributions were determined using Malvern/MS 3000 laser grain size analyzer. The samples
used for TOC and TN determination were decalcified with 10% HCl and rinsed repeatedly with deionized
water. Then, ~3-5 mg dried and ground samples were analyzed using the EURO EA 3000 elemental
analyzer.

Terrestrial plant remains at different depths in DLC1819 core were used for accelerator mass

spectrometry (AMS) [14]C measurements at Beta Analytic Inc., U.S.A. (Table 1). The uppermost sediments
were used for the radiometric dating by measuring the activity of [137]Cs as a function of depth. The
samples of the uppermost sediments at 0.5 cm intervals were dried and ground to less than 100 mesh and
then loaded into the 5 mL cylindrical PVC tube. Radio-activities measurement of [137]Cs was measured by
Spectrum Analysis System consisting of a high-purity germanium well detector produced by American
EG& G Ortec Company, Ortec 919 spectrum controller, and IBM microcomputer with a 16 K channel
multichannel analyzer.
**4 Results**
**4.1 Chronology**

A significant increase in [137]Cs activities occurred at approximately 71 cm, which could be attributed

to the onset of rising concentrations of [137]Cs in the Northern Hemisphere (NH) at 1952 CE (Fig. 3a). The
distinct peak at the depth of 64 cm was taken as the 1963 CE global fallout maximum (Pennington et al.,
1973) (Fig. 3a). We established the age-depth model of DLC1819 based on the 2 [137]Cs ages and 9
radiocarbon ages by the Bacon 2.2 procedure in R 3.2 software using the Bayesian method (Blaauw and
Christen, 2011) (Fig. 3b). Chronology results show that DLC1819 core covers the past 840 years with an
approximate average sedimentation rate of 0.9 cm/yr.
**4.2 TOC, TN, C/N, MS, Grainsize**

TOC and TN results show broadly similar changes and vary between 1.14% and 8.33%, and between

0.09% and 0.59%, respectively (Fig. 2). C/N ratios fluctuate between 7.95 and 18.42 with an average of
10.88 and the values of C/N ratios exceed 10 at the depth of 440-650 cm, 330-400 cm, 230-300 cm, and



130-180 cm (Fig. 2). MS values of DLC1819 core vary between 5.90 and 41.89×10$^{-8}$ m$^3$/Kg with an
average of 20.66×10$^{-8}$ m$^3$/Kg (Fig. 2). The silt fraction values fluctuate from 43.64% to 88.50% with an
average of 71.91% while the variation of clay fraction has an opposite trend to that of silt, and its values
vary from 11.84% to 54.26% with an average of 25.97% (Fig. 2). Sand fraction only accounts for 2.43%
of the total grain size on average. Generally, the higher values of the MS and silt fraction and the lower
values of the clay fraction correspond to the higher C/N ratios and vice versa (Fig. 2).
**4.3 Humidity Index reconstruction**
Lake Dalongchi is a typical small lake with a surface area of ~1.4 km$^2$. The drainage basin is
surrounded by high mountains and has a steep headwall. Thus, it possesses sediments that preserve a
highly sensitive record of past watershed material input changes. During the humid/dry period
represented by high/low lake level and enlarged/reduced lake area, exogenous materials containing
magnetic minerals, coarse grain components, and terrestrial plants were poorly/easily transported to the
site from which we took the core due to the long/short distance from the lakeshore and
reduction/intensified erosion in the basin (Fig. 4). Terrestrial plants usually have C/N ratios of more than
20 (Meyers, 1994, 2003), so the increased C/N ratios reflect the input amount of allochthonous organic
matter. Thus, the high susceptibility, silt content, and C/N ratios indicate a dry climate and vice versa
(Fig. 4).
Accordingly, multiple proxies, such as C/N ratios, MS, silt, and clay fractions, were synthetically
employed to reconstruct the Humidity Index (HI) in Lake Dalongchi region over the past millennium
(Fig. 5). As the high values of C/N ratios, MS, and silt contents, and low values of clay content reflect
the arid climatic environment, the first three records multiplied by (-1) and clay content were normalized
to a Z-score (Figs. 5b, c, d, e). Then the HI was derived from the average of the normalized standard Z-
scores. Positive and negative Z-scores indicate dry and wet climatic conditions (Fig. 5a).
**5 Discussion**
**5.1 The humidity changes over the last millennium**
The reconstructed HI shares good consistency with the instrumental relative humidity records over
the past 60 years ($r$ = 0.298*) from the nearby Bayanbuluk meteorological station (Fig. 6a), verifying the
reliability of the humidity reconstruction. The HI changes show that the climate was dry during the MWP
(1180-1420 CE) and CWP (1920-2018 CE), and wet during the LIA (1420-1920 CE) (Fig. 6b). This



multi-centennial climate pattern is in agreement with the hydroclimatic patterns revealed by numerous
studies in ACA (Chen et al., 2006; Song et al., 2015; Lan et al., 2018; Lan et al., 2019; Zhao et al., 2009;
He et al., 2013; Ma and Edmunds, 2006; Gates et al., 2008) (Figs. 6c, d, e, f, g). Previous studies indicate
the negative phase of the NAO and AMO during the LIA favors increasing precipitation in ACA (Chen
et al., 2019b; Chen et al., 2015; Lan et al., 2018; Aichner et al., 2015; Chen et al., 2006; Chen et al.,
2016). During the MWP, the positive phase of the NAO with enhanced pressure between the Azores High
and the Icelandic Low would lead to strengthened zonal flow, and the axis of maximum moisture
transport and preferred storm track extend to the north and east (Trouet et al., 2009). In contrast, the axis
of maximum moisture transport and preferred storm track migrated southwards when the NAO was in a
negative phase during the LIA (Hurrell, 1995). A general cold (warm) phase of the AMO corresponds to
the negative (positive) NAO phase during the LIA (MWP) (Wang et al., 2017; Ortega et al., 2015),
leading to a weaker (strong) upper-level jet stream intensity and further resulting in development
(recession) of the through-ridge system, consequently contributing to the increased (decreased)
precipitation in ACA (Chen et al., 2019b). Thus, the multi-centennial behavior in our reconstruction
might be related to the influence of the NAO and AMO on hydroclimate changes during in MWP to LIA.

Notably, distinct from other records in ACA (Chen et al., 2006; Ma and Edmunds, 2006; Gates et

al., 2008; He et al., 2013), our reconstruction shows dramatic and unstable multidecadal to century-scale
climatic variability, especially within the LIA (Fig. 6b). Four wet episodes with the sharp high HI values
were recorded in the 1420-1470, 1550-1600, 1650-1720, 1800-1920 CE, and three dramatic dry periods
with the low HI values were recorded in the 1470-1550, 1600-1650, 1720-1800 CE during the LIA (Fig.
6b). Continuous Wavelet Transform (CWT) of HI exhibits a significant century-scale dominant
oscillation ranging from 88 to 157 years, which is nearly throughout the entire time series and prominent
in 1450-1800 CE, as well as a strong 50~65-year multidecadal oscillation at a 95% confidence level
relative to the red noise spectrum (Fig. 7a).
**5.2 The role of the Gleissberg cycle**

There is a strong link and inverse relationship between the HI and TSI (Fig. 7b). The HI increased

significantly and reached its peak during the several grand solar minimums (Wolf, Spörer, Maunder,
Dalton), whereas the humidity decreased rapidly in the maximum solar activity period (Fig. 7b). The
Wavelet Coherence (WTC) spectrum between HI and TSI shows a strong correlation and anti-phase



pattern (Fig. 7c). Periodicities of significant coherence for HI and TSI occurred at 88 to 146 years,
particularly from 1400 CE to the present. Arrows in the significant coherence spectral area point almost
entirely to the left, implying the persistent negative correlation of HI and TSI (Fig. 7c). The WTC result
confirms that the persistent 88-157-year cycle of HI in CWT is associated with solar activity (Figs. 7a,
c). The cycle of 88-146 years should be attributed to the century-scale solar cycle of Gleissberg
(Gleissberg, 1958; Gleissberg, 1965; Ogurtsov et al., 2015). Shindell et al. examined the climate response
to the solar forcing at the Maunder Minimum and indicated that even relatively small solar activity might
play a primary role in century-scale climate change in NH (Shindell et al., 2001). The possible solar
contribution of the Gleissberg century cycle to climate changes over at least the last millennium is mainly
concentrated in the North Atlantic region (Moffa-Sánchez et al., 2014; Ogurtsov et al., 2002b; Ogurtsov
et al., 2002a; Ogurtsov et al., 2015). Our reconstruction provides strong evidence for negative link
between the Gleissberg solar cycle and humidity changes at century timescales.

The sediment records from Lake Manas in ACA showed a negative relationship between

temperature and moisture variation on century timescales (Song et al., 2015). Moreover, the inverse
relationship between lake-level proxies and precipitation in Qaidam Basin suggested that increased
evaporation rather than decreased inflow was more responsible for lowered lake levels (Zhao et al., 2009).
Therefore, a solar forcing might weaken the influence of mid-latitude precipitation and play an important
role in regulating effective humidity in ACA at century timescales by controlling temperature and
evaporation.
**5.3 Linkage to ENSO**

Paleoclimatic proxies and historical records suggest that the El Niño-Southern Oscillation (ENSO)

has long-term variability in amplitude and frequency on multidecadal to centennial timescales (Yeh and
Kirtman, 2007; D'arrigo et al., 2005; Li et al., 2011; Mann et al., 2000). On the multi-centennial timescale
(i.e., MWP to LIA), the HI and ENSO variance (Li et al., 2011) show a similar trend in amplitude changes
(Fig. 7d). The amplitudes of HI and ENSO variance both show a distinctly increasing trend during the
LIA and maintain a relatively high level from ~1650 to 1950 CE (Fig. 7d). This trend suggests that the
relatively humid environment and unstable hydroclimate in ACA may be associated with the increase in
ENSO variance amplitude (Li et al., 2011) and more frequent ENSO events (Rustic et al., 2015) during
the LIA. The results here support previous studies showing that ENSO might affect hydroclimate



variability in ACA at multi-centennial timescales with La Niña-like (EI Niño-like) conditions during the
MWP (LIA) (Chen et al., 2019a; Chen et al., 2015).
On the multidecadal timescales, however, the WTC between the HI and ENSO variance shows a
robust negative phase relationship (Fig. 7e). In particular, the WTC result shows that the HI has a similar
quasi-regular cycle of ENSO variance from 82-90-yr during the MWP to 50-60-yr thereafter (Li et al.,
2011), which reveals the potential modulation of these quasi-regular cycles of ENSO variance to extreme
humidity oscillations at the multidecadal timescales in ACA (Fig. 7e). Daily observational precipitation
and National Center for Environmental Prediction (NCEP) reanalysis data suggest that the low-level
water vapor fluxes from the Indian Ocean, transported along the eastern periphery of the Tibetan Plateau,
are the most important factor leading to rainstorms in ACA (Huang et al., 2017). Thus, the quasi-regular
periodic variation in ENSO amplitude is likely to influence the extreme hydroclimate at multidecadal
scales in ACA over the past millennium by altering the meridional circulation conditions (Huang et al.,
2017). However, the mechanisms for the different timescales between the ENSO amplitude and
hydroclimate changes in ACA require further exploration through high-resolution records and simulation
experiments.
**6 Conclusions**
We present the Humidity Index (HI) in ACA over the past millennium based on the ~1.8-year high-
resolution multiproxy records from Lake Dalongchi in the central Tianshan Mountains. Our results reveal
dramatic and unstable multidecadal to century-scale humidity oscillations over the last millennium,
especially within the LIA, which is distinct from other records of ACA. Our findings emphasize that the
Gleissberg solar cycle and quasi-regular period of ENSO amplitude play critical roles in controlling the
effective humidity in ACA at century and multidecadal timescales, respectively. However, high-
resolution records at different time scales and climate model simulations are still needed to improve our
understanding of the physical mechanisms of the links between solar irradiance and ocean-atmosphere
modes and how their coupling affects moisture variation in ACA.
**Data availability**
The reconstructed Humidity Index in this study are submitted to the datasets of the 4TU Center for
Research Data, which can be temporarily available at https://figshare.com/s/66270515b62cb166d8c9.
**Author contributions**



SF and XL conceived this study, carried out the laboratory analysis and data interpretation, and
wrote the manuscript; FS and YL performed the data analysis; XM and JW participated in the retrieval
of the sediment core and sampling. All authors discussed the results and commented on the manuscript.
**Competing interests**
The authors declare that they have no conflict of interest.
**Financial support**
This research has been supported by the National Key Research and Development Program of China
(Grant No. 2018YFA0606400), the National Natural Science Foundation of China (Grant No. 41907375),
and the Basic Research Program of the Institute of Hydrogeology and Environmental Geology CAGS
(Grant No. SK202007).

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





**Table 1.** Accelerator mass spectrometry (AMS) $^{14}$C dating results of DLC1819 core.

| No. | Lab. No | Sample ID | Composite depth (cm) | Analyzed material | δ$^{13}$C (‰) | $^{14}$C age /a BP | Calendar age/CE |
|-----|---------|-----------|----------------------|-------------------|---------------|--------------------|-----------------|
| 1 | Beta - 514897 | DLC-1-1-61 | 69 | Wood | -24.9 | 90+/-30 | 1870-1928 |
| 2 | Beta - 507553 | DLC-1-1-135 | 143 | Wood | -27.4 | 170+/-30 | 1721-1818 |
| 3 | Beta - 507554 | DLC-1-2-82 | 210 | Wood | -31.4 | 230+/-30 | 1635-1684 |
| 4 | Beta - 507555 | DLC-1-2-155 | 283 | Wood | -26.1 | 340+/-30 | 1470-1640 |
| 5 | Beta - 507556 | DLC-1-3-61 | 324 | Wood | -28.1 | 430+/-30 | 1420-1498 |
| 6 | Beta - 507557 | DLC-1-3-119 | 382 | Wood | -22.3 | 440+/-30 | 1416-1490 |
| 7 | Beta - 514898 | DLC-1-3-131 | 394 | Wood | -23.7 | 370+/-30 | 1446-1528 |
| 8 | Beta - 514901 | DLC-3-5'-37 | 585 | Wood | -23.6 | 670+/-30 | 1274-1320 |
| 9 | Beta - 542591 | DLC2019-1-5-76 | 625 | Wood | -23.1 | 800+/-30 | 1184-1275 |


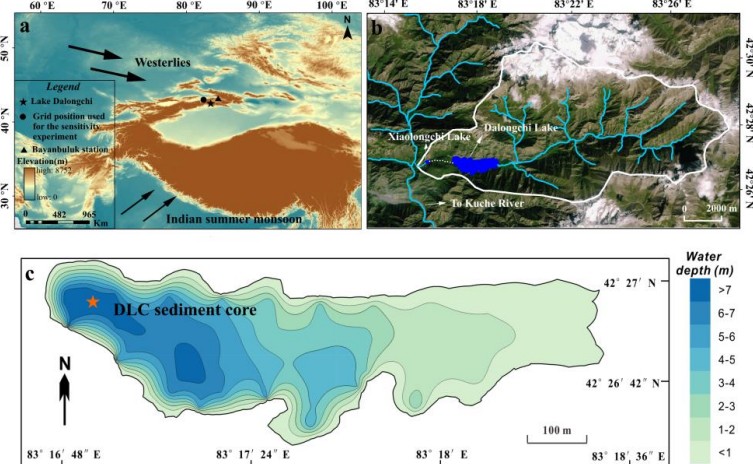


**Figure 1.** Maps of the study site. (a) The location of Lake Dalongchi. The map was generated by ArcMap
10.2 software (ESRI, USA, http://www.esri.com). (b) Watershed map of the study site. (c) Bathymetric
contour map of Lake Dalongchi with the coring site.



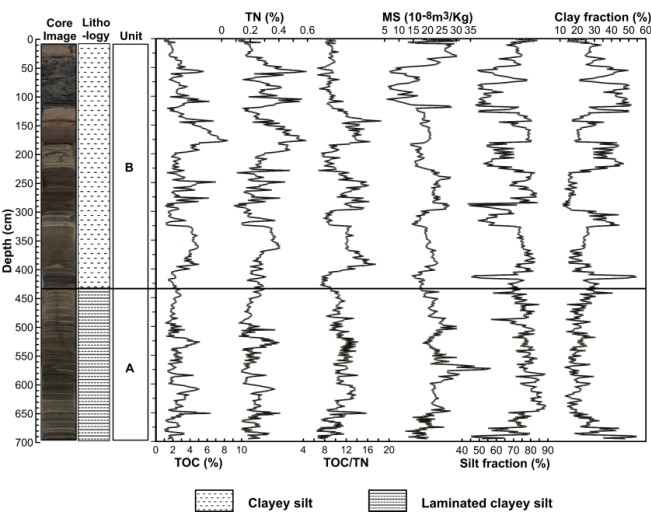

Figure 2. Sedimentary lithology and multi-proxy variation versus depth for the composite DLC1819

core.

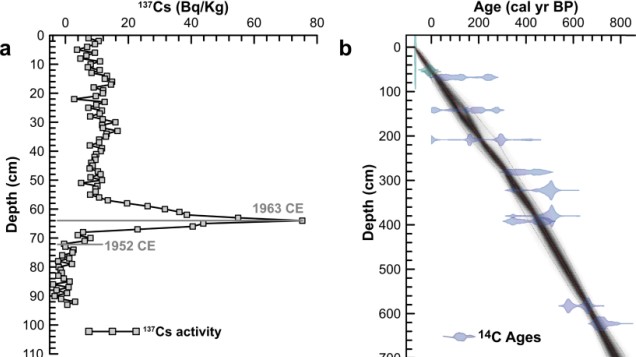

Figure 3. Age-depth model for the DLC1819 core of Lake Dalongchi. (a) The [137]Cs activity versus depth

in the uppermost sediments. (b) Bayesian age-model for the calibrated ages. Black dots indicate the 95%

probability intervals of the model.



375

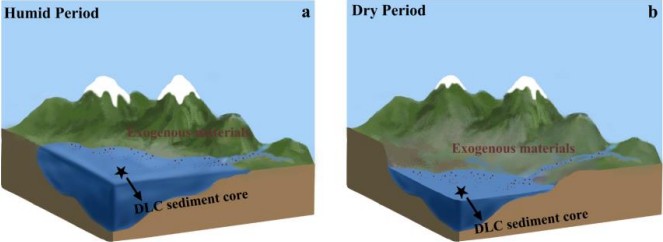

**Figure 4.** Cartoon schematic diagram illustrating a simple explanation of the lacustrine depositional

process in Lake Dalongchi region. Lake level condition and the transport process of exogenous materials

in the humid period (a) and the dry period (b).

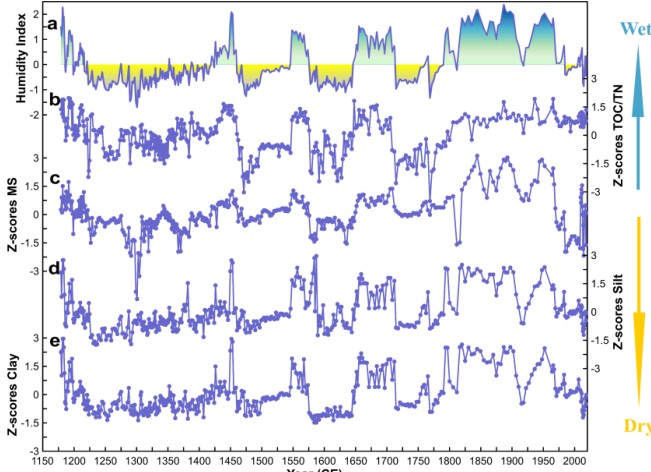

379

**Figure 5.** Humidity Index reconstruction (a) based on the Z-scores of total organic carbon/total nitrogen

(TOC/TN) ratios (b), the Z-scores of magnetic susceptibility (MS) (c), the Z-scores of silt fraction (d),

and the Z-scores of clay fraction variation (e) in the DLC1819 core in Lake Dalongchi.

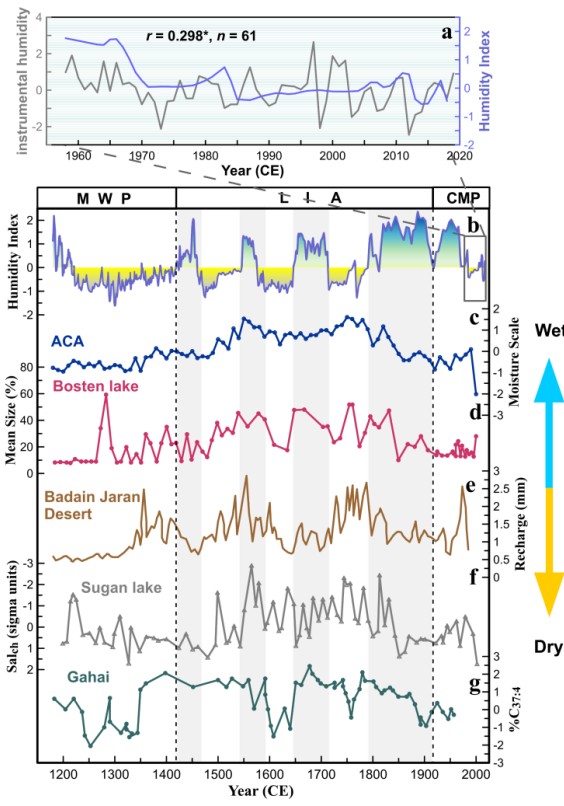

**Figure 6.** Humidity Index (HI) of Lake Dalongchi and comparison to other records in ACA over the last millennium. (a) Comparison between the HI (blue line) and the standardized instrumental effective humidity recorded by Bayanbuluk meteorological station (grey line). * represents the 0.05 significance level. (b) The reconstructed HI for the past millennium (1180-2018 CE). (c) The synthesized moisture curve over the last millennium in ACA (Chen et al., 2010). (d) Variations in the mean grain size along the BST04H core in Bosten Lake (Chen et al., 2006). (e) The unsaturated recharge history in Badian Jaran (Gates et al., 2008; Ma and Edmunds, 2006). (f) Chironomid inferred salinity (Salch) in SG03I of Sugan Lake (Chen et al., 2009). (g) %C$_{37:4}$ from Lake Gahai (He et al., 2013). Light grey bars highlight intervals of increased humidity in Lake Dalongchi during the LIA.



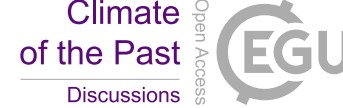

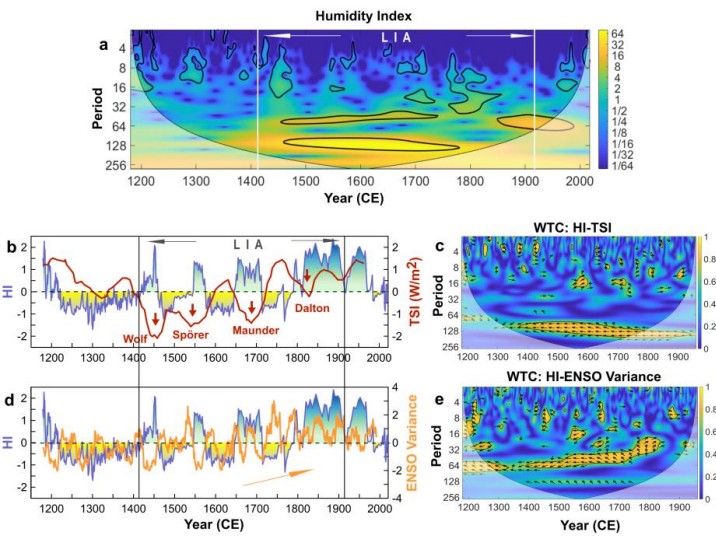

**Figure 7.** The wavelet analysis of the HI and the relationship between the HI and the TSI or ENSO. (a) Continuous wavelet (CWT) power spectrum of the HI. The irregular thick black contour represents the 95% confidence level against red noise and the thin curved black solid line is the cone of influence (COI) (Grinsted et al., 2004). (b) A comparison between the HI and the reconstructed total solar irradiance (TSI) (Bard et al., 2000). Wolf, Spörer, Maunder, and Dalton represent several grand solar minimums that occurred during the LIA. (c) The wavelet coherence (WTC) result between the HI and the TSI (Bard et al., 2000). (d) As (b), but for ENSO variance reconstruction (Li et al., 2011). Orange arrows represent the enhanced ENSO amplitude trend. (e) As (c), but for ENSO variance (Li et al., 2011). The 95% confidence level against red noise is shown as an irregular thick black contour. The black arrows illustrate the relative phase relationship: arrows pointing right are in-phase and those pointing left are anti-phase (Grinsted et al., 2004).