# Peer review of "Humidity changes and possible forcing mechanisms over"

_Climate of the Past, 2021_

## Author Comment (AC1)

**Author response to Anonymous Referee #1**

**Article ID: cp-2021-137**

**General comments:**

The instability of humidity variation, especially on decadal to multi-decadal timescales, has a profound impact on human welfare in arid central Asia (ACA). However, it is still uncertain whether the regional hydrological evolution was controlled by external driving or internal driving. In this study, the authors provided a ~1.8-year high-resolution humidity record spinning the past 840 years from Lake Dalongchi. Based on this record, a dry Medieval Warm Period (MWP) and a wet, unstable Little Ice Age (LIA) was determined. Moreover, they suggested that the climate instability during the LIA was controlled by Gleissberg solar cycle and ENSO on centennial and multi-decadal timescales, respectively. Such high-resolution record is very rare in the ACA. This manuscript provides critical insights into regional climate change history, although the results still need to be further corroborated. Therefore, I suggest that the manuscript can be accepted for publication after a minor revision.

We are very grateful to the reviewer for your careful review and helpful comments on this paper. These suggestions are conductive to revise and improve our manuscript. Accordingly, we have prepared detailed point-by-point responses below. Our responses to the comments have been made in blue. Line numbers and revised figure numbers that refer to the changes in the revised manuscript have been marked in red.

**Major comments:**

1. A previous study discussed climate instability in Northwestern China during the LIA, which proposed that the instability of North Atlantic Oscillation (NAO) was a major driving factor (Chen et al., 2019). In this manuscript, the authors emphasized the influences of solar forcing and ENSO on climate instability during the LIA, but neglected the influences of NAO. It would make the conclusions more complete by adding the related discussion in the manuscript.

Chen, J., Liu, J., Zhang, X., Chen, S., Huang, W., Chen, J., Zhang, S., Zhou, A., and Chen, F. Unstable Little Ice Age climate revealed by high-resolution proxy records from northwestern China, Climate Dynamics, 2019, 53, 1-10.

Good points. Thanks for the comment. NAO and AMO are important factors in influencing the hydroclimate changes of ACA during the past millennium. As suggested, we have added a section "5.3.1 The influence of NAO and AMO" to provide more

corresponding discussion through comparing our reconstruction with the reconstructed NAO and AMO (Ortega et al., 2015; Wang et al., 2017). The results show that the relationship between HI and NAO, and AMO is ambiguous on multidecadal timescales, and the Wavelet Coherence (WTC) results also show their relationships are nonpersistent. But on the multi-centennial timescales, our reconstruction reveals the dominant dry climate conditions during the Medieval Warm Period (MWP) and humid climate conditions during the Little Ice Age (LIA), which is related to the generally positive and negative phases exhibited in the NAO and AMO. This is in agreement with the previous studies (Chen et al., 2006; Lan et al., 2018; Chen et al., 2019b; Chen et al., 2019a). The relevant discussions added in the manuscript make the conclusions more complete. (Lines 237-264, Fig. 6 and Fig. S2 in the Supplement)

2. During 1400-1800 CE, the variation of humidity index was negatively correlated with that of ENSO variance (Fig. 7d). However, the humidity index showed consistent variation with ENSO variance during 1800-1950 CE. How to explain the complicated relationship between humidity index and ENSO variance? More discussion is needed in here.

Thanks for the comment. There is a negative phase relationship between the HI and ENSO variance at multidecadal timescales. However, this relationship can only be revealed by the WTC results, rather than the two datasets, because the original HI contains a variety of signals at different timescales. To solve such a problem, we further performed the ensemble empirical mode decomposition (EEMD), a new noise-assisted data analysis method (Huang et al., 1998; Wu and Huang, 2009), to extract the multidecadal signals of the HI. Interestingly, the extracted multidecadal component of the HI exhibits better inverse relationship with the ENSO variance almost throughout the entire time series (*please see the following Figure 1*), which is in line with the WTC results. More discussions were added in the revised manuscript. (Lines 358-377, Fig. 7g and Fig. S3b in the Supplement)

*Figure 1. The comparison between the multidecadal component of HI extracted by EEMD and ENSO variance (Li et al., 2011).*

3. Figure 6 shows the comparison of humidity records from Lake Dalongchi with other records. However, the authors only use one sentence to describe it (Lines 146-149). It would be better to add more statements.

Thanks for the comment. We have added more discussions about the comparison of our humidity records with other records. (Lines 207-232)

**Minor comments**

1. In Line 140, the authors stated that "Positive and negative Z-scores indicate dry and wet climatic conditions (Fig. 5a)." But this statement is opposite to the contents of Figure 5.

Thanks! We have revised the manuscript accordingly. (Line 198)

2. The authors used bacon 2.2 to establish the age-depth model. However, the latest version is bacon 2.5.7, which used new calibration curve.

Thanks for the reminder. We have used bacon 2.5.7 version to establish the age-depth model in the revised manuscript and updated all figures based on new chronology model.

3. The statement of "at different timescales" should be changed to "on different timescales". This issue should be revised throughout the whole manuscript.

Thanks! We have revised the whole manuscript accordingly.

4. Line 89: Is "mass susceptibility" supposed to "magnetic susceptibility"?

Yes. Thank you. We have revised it to "magnetic susceptibility". (Line 130)

5. Line 119, 120: Change "fraction" to "percentage".

Thanks. We have revised the manuscript accordingly. (Line 167, 168)

6. Line 133: Change "high susceptibility" to "high MS values".

Thanks. We have revised it. (Line 191)

7. Line 143: Given that the correlation coefficient between humidify index and instrumental data is only 0.298, it's inaccurate to state it as "good consistency".

Thanks for your reminder. We appreciate this comment. We have revised our statement in the manuscript as follows: "There is a generally positive correlation (r = 0.298\*) between the reconstructed HI and the instrumental relative humidity records

over the past 60 years from the nearby Bayanbuluk meteorological station at the 0.05 significant level, verifying the reliability of the humidity reconstruction." (Lines 200-202).

8. Line 245-247: The DOI of this paper was missed.

Thanks. We have revised it. (Lines 406-408)

9. Line 358: The "." should be removed from "10.1029/2009GL040951.,".

Thanks. We have revised it. (Line 689)

**References**

- Chen, F., Huang, X., Zhang, J., Holmes, J. A., and Chen, J.: Humid Little Ice Age in arid central Asia documented by Bosten Lake, Xinjiang, China, Science in China Series
   D: Earth Sciences, 49, 1280-1290, 10.1007/s11430-006-2027-4, 2006.
- Chen, F., Chen, J., Huang, W., Chen, S., Huang, X., Jin, L., Jia, J., Zhang, X., An, C., Zhang, J., Zhao, Y., Yu, Z., Zhang, R., Liu, J., Zhou, A., and Feng, S.: Westerlies Asia and monsoonal Asia: Spatiotemporal differences in climate change and possible mechanisms on decadal to sub-orbital timescales, Earth Science Reviews, 192, 337-354, 10.1016/j.earscirev.2019.03.005, 2019a.
- Chen, J., Liu, J., Zhang, X., Chen, S., Huang, W., Chen, J., Zhang, S., Zhou, A., and Chen,
  F.: Unstable Little Ice Age climate revealed by high-resolution proxy records from northwestern China, Climate Dynamics, 53, 1-10, 10.1007/s00382-019-04685-5, 2019b.
- Huang, N. E., Shen, Z., Long, S. R., Wu, M. C., Shih, H. H., Zheng, Q., Yen, N.-C., Tong, C.
  C., and Liu, H. H.: The empirical mode decomposition and the Hilbert spectrum for nonlinear and non-stationary time series analysis, Proceedings of the Royal Society of London, 454, 903-995, 10.1098/rspa.1998.0193, 1998.
- Lan, J., Xu, H., Sheng, E., Yu, K., Wu, H., Zhou, K., Yan, D., Ye, Y., and Wang, T.: Climate changes reconstructed from a glacial lake in High Central Asia over the past two millennia, Quaternary International, 487, 43-53, 10.1016/j.quaint.2017.10.035, 2018.
- Li, J., Xie, S.-P., Cook, E. R., Huang, G., D'Arrigo, R., Liu, F., Ma, J., and Zheng, X.-T.: Interdecadal modulation of El Niño amplitude during the past millennium, Nature Climate Change, 1, 114-118, 10.1038/NCLIMATE1086, 2011.
- Ortega, P., Lehner, F., Swingedouw, D., Masson-Delmotte, V., Raible, C. C., Casado, M., and Yiou, P.: A model-tested North Atlantic Oscillation reconstruction for the past millennium, Nature, 523, 71-74, 10.1038/nature14518, 2015.
- Wang, J., Yang, B., Ljungqvist, F. C., Luterbacher, J., Osborn, Timothy J., Briffa, K. R., and Zorita, E.: Internal and external forcing of multidecadal Atlantic climate variability

over the past 1,200 years, Nature Geoscience, 10, 512-517, 10.1038/NGEO2962, 2017.

Wu, Z. and Huang, N. E.: Ensemble Empirical Mode Decomposition: A Noise-Assisted Data Analysis Method, Advances in Adaptive Data Analysis, 1, 1-41, 2009.

---

## Author Comment (AC2)

**Author response to Anonymous Referee #2**

**Article ID: cp-2021-137**

**General comments:**

I believe high-resolution hydroclimatic records during the last millennium in Tienshan Mts will be welcomed by both paleoclimatologists and archeologists. This study made a good attempt. In general, the manuscript is properly organized and well written. The scientific topic is significant and main conclusions are convincing. It is suitable for the scope of this journal. Therefore, I recommend acceptance of the manuscript for publication after some minor revisions.

We thank the reviewer very much for reviewing the manuscript. All suggestions are careful and insightful, which will help us improve the manuscript. Accordingly, we have prepared detailed point-by-point responses below. Our responses to the comments have been made in *blue*. Line numbers that refer to the changes in the revised manuscript version have been marked in *red*.

1. There are still some records suggesting a humid MWP around Tianshan Mts (e.g., Zhang et al., 2009[doi: 10.1029/2009gl037375]). Could you please provide some discussions?

Thanks for the comment. Yes, a humid climate during the MWP was documented by the tree ring of Sabina przewalskii Kom, the Lop Nur, and the Daxigou profile in the Tianshan Mountains, etc. in the earlier studies (Zhang et al., 2009; Ma et al., 2008; Zhang et al., 2003). However, a growing body of studies based on the variety of paleoclimate records show a general climate pattern of a relatively dry MWP and a wet LIA (Chen et al., 2006; Song et al., 2015; Lan et al., 2018; Lan et al., 2019; Zhao et al., 2009; He et al., 2013; Ma and Edmunds, 2006; Gates et al., 2008; Rousseau et al., 2020; Chen et al., 2015; Chen et al., 2010; Chen et al., 2019). We have provided relevant discussions in sections of "Introduction" and "Discussion" according to the above two respects. (Lines 49-58, 207-212)

2. Previous studies have already proposed "an unstable hydroclimate during the LIA over the ACA" (e.g., Chen et al., 2009 [doi: 10.1007/s11434-009-0201-8], 2019[doi: 10.1007/s00382-019-04685-5]) on multi-decadal to centennial timescales, which should not be neglected in the relevant discussion part.

Good suggestion! The research of the unstable hydroclimate during the LIA is an important reference, but it is not clear how the specific unstable wet and dry climate

fluctuated during the LIA. Our reconstruction provides new evidence for the unstable hydroclimate variability during the LIA. More relevant discussions have been added in "section 5.2". (Lines 220-232)

3. In "5.3 Linkage to ENSO", the significance of ENSO variance for hydroclimate in ACA, or Asia, should be firstly pointed out. It is also notable that the referred work (Huang et al., 2017) could not be used as evidence for the influence of ENSO variance on extreme rainfall events in this region - It focuses on ENSO itself rather than ENSO variance.

Thanks for the comment. The significance of ENSO variance for hydroclimate in ACA has been added in the discussion in "section 5.3.3 Linkage to ENSO" of the revision. The influence of ENSO on the extratropical climate has been shown to be modulated by ENSO variance at multidecadal timescales, and the calculated 31-year running correlations between the reconstructed ENSO variance and other records of ENSO teleconnections shows that the ENSO teleconnection is robust over Central Asia during the past seven centuries only except for the Maunder minimum (Li et al., 2013). (Lines 338-339, 360-366)

Moreover, the ENSO variance is the calculated 21-yr running biweight variance derived by the ENSO variability, reflecting changes in ENSO itself (Li et al., 2011). In other words, the changes of ENSO itself contribute to the multidecadal-timescale ENSO variance. In the manuscript, we suggest that the ENSO variance effect on the hydroclimate changes in ACA might be through modulating the extreme precipitation. Previous studies indicate that the water vapor from the Arabian Sea may be transported to the Xinjiang region and cause heavy precipitation, although the water vapor fluxes mostly come from the west transported by the prevailing westerlies (Huang et al., 2015; Huang et al., 2013). Observational reanalysis data show that water vapor in ACA also comes from the Indian Ocean and cause heavy precipitation, which gives us a good theoretical and data support, although the driving mechanisms of ENSO variance for the hydroclimate changes in ACA require further exploration through high-resolution records and simulation experiments. We have added more relevant discussions in the section "Linkage to ENSO" of the revised manuscript. (Lines 364-377)

**Technical comments:**

1. The contour lines should be further smoothed in Figure 1c.

Thank you. We have further smoothed the contour lines in Figure 1c.

2. In Figure 3b, I know what you mean, but where are the "black dots"?

Thanks for this comment. We have revised "black dots" to "Black dotted lines".

3. It would be better to use "centennial" timescale when "multi-decadal" timescale was used.

Thanks for the comment. Yes, the "centennial" is a good choice, but it was also usually referred to as "multi-centennial". The aim we used "century" in this manuscript is to correspond to cycles of solar activity from 88 to 146 years because the Glasberg cycle is also known as the century-type cycle (Gleissberg, 1958; Ogurtsov et al., 2002). Thus, the "multi-decadal, century, and multi-centennial" were used to correspond different forcing at different timescales.

4. P5L140, "Positive and negative Z-scores indicate dry and wet climatic conditions". It seems to me that positive Z-scores indicate dry conditions, and vice versa. Please revise it.

Thanks! We have revised the manuscript accordingly. (Line 198)

**References.**

Chen, F., Huang, X., Zhang, J., Holmes, J. A., and Chen, J.: Humid Little Ice Age in arid central Asia documented by Bosten Lake, Xinjiang, China, Science in China Series D: Earth Sciences, 49, 1280-1290, 10.1007/s11430-006-2027-4, 2006.

Chen, F., Chen, J., Holmes, J., Boomer, I., Austin, P., Gates, J. B., Wang, N., Brooks, S. J., and Zhang, J.: Moisture changes over the last millennium in arid central Asia: a review, synthesis and comparison with monsoon region, Quaternary Science Reviews, 29, 1055-1068, 10.1016/j.quascirev.2010.01.005, 2010.

Chen, F., Chen, J., Huang, W., Chen, S., Huang, X., Jin, L., Jia, J., Zhang, X., An, C., Zhang, J., Zhao, Y., Yu, Z., Zhang, R., Liu, J., Zhou, A., and Feng, S.: Westerlies Asia and monsoonal Asia: Spatiotemporal differences in climate change and possible mechanisms on decadal to sub-orbital timescales, Earth Science Reviews, 192, 337-354, 10.1016/j.earscirev.2019.03.005, 2019.

Chen, J., Chen, F., Feng, S., Huang, W., Liu, J., and Zhou, A.: Hydroclimatic changes in China and surroundings during the Medieval Climate Anomaly and Little Ice Age: spatial patterns and possible mechanisms, Quaternary Science Reviews, 107, 98-111, 10.1016/j.quascirev.2014.10.012, 2015.

Gates, J. B., Edmunds, W. M., Ma, J., and Sheppard, P. R.: A 700-year history of groundwater recharge in the drylands of NW China, The Holocene, 18, 1045-1054, 10.1177/0959683608095575, 2008.

Gleissberg, W.: The eighty-year sunspot cycle, J. Br. Astron. Assoc., 68, 148-152, 1958.

He, Y., Zhao, C., Wang, Z., Wang, H., Song, M., Liu, W., and Liu, Z.: Late Holocene coupled moisture and temperature changes on the northern Tibetan Plateau,

Quaternary Science Reviews, 80, 47-57, 10.1016/j.quascirev.2013.08.017, 2013.

Huang, W., Feng, S., Chen, J., and Chen, F.: Physical Mechanisms of Summer Precipitation Variations in the Tarim Basin in Northwestern China, Journal of Climate, 28, 3579-3591, 10.1175/jcli-d-14-00395.1, 2015.

Huang, W., Chen, F., Feng, S., Chen, J., and Zhang, X.: Interannual precipitation variations in the mid-latitude Asia and their association with large-scale atmospheric circulation, Chinese Science Bulletin, 58, 3962-3968, 10.1007/s11434-013-5970-4, 2013.

Lan, J., Xu, H., Sheng, E., Yu, K., Wu, H., Zhou, K., Yan, D., Ye, Y., and Wang, T.: Climate changes reconstructed from a glacial lake in High Central Asia over the past two millennia, Quaternary International, 487, 43-53, 10.1016/j.quaint.2017.10.035, 2018.

Lan, J., Xu, H., Yu, K., Sheng, E., Zhou, K., Wang, T., Ye, Y., Yan, D., Wu, H., Cheng, P., Abuliezi, W., and Tan, l.: Late Holocene hydroclimatic variations and possible forcing mechanisms over the eastern Central Asia, Science China, 62, 1288-1301, 10.1007/s11430-018-9240-x, 2019.

Li, J., Xie, S.-P., Cook, E. R., Huang, G., D'Arrigo, R., Liu, F., Ma, J., and Zheng, X.-T.: Interdecadal modulation of El Niño amplitude during the past millennium, Nature Climate Change, 1, 114-118, 10.1038/NCLIMATE1086, 2011.

Li, J., Xie, S.-P., Cook, E. R., Morales, M. S., Christie, D. A., Johnson, N. C., Chen, F., D'Arrigo, R., Fowler, A. M., Gou, X., and Fang, K.: El Niño modulations over the past seven centuries, Nature Climate Change, 3, 822-826, 10.1038/nclimate1936, 2013.

Ma, C., Wang, F., Cao, Q., Xia, X., Li, S., and Li, X.: Climate and environment reconstruction during the Medieval Warm Period in Lop Nur of Xinjiang, China, Science Bulletin, 53, 3016-3027, 10.1007/s11434-008-0366-6, 2008.

Ma, J. and Edmunds, W. M.: Groundwater and lake evolution in the Badain Jaran Desert ecosystem, Inner Mongolia, Hydrogeology Journal, 14, 1231-1243, 10.1007/s10040-006-0045-0, 2006.

Ogurtsov, M. G., Nagovitsyn, Y. A., Kocharov, G. E., and Jungner, H.: Long-Period Cycles of the Sun's Activity Recorded in Direct Solar Data and Proxies, Solar Physics, 211, 371-394, 10.1023/A:1022411209257, 2002.

Rousseau, M., Demory, F., Miramont, C., Brisset, E., Guiter, F., Sabatier, P., and Sorrel, P.: Palaeoenvironmental change and glacier fluctuations in the high Tian Shan Mountains during the last millennium based on sediments from Lake Ala Kol, Kyrgyzstan, Palaeogeography, Palaeoclimatology, Palaeoecology, 558, 10.1016/j.palaeo.2020.109987, 2020.

Song, M., Zhou, A., Zhang, X., Zhao, C., He, Y., Yang, W., Liu, W., Li, S., and Liu, Z.: Solar

imprints on Asian inland moisture fluctuations over the last millennium, The Holocene, 5, 1935-1943, 10.1177/0959683615596839, 2015.

Zhang, Q.-B., Cheng, G., Yao, T., Kang, X., and Huang, J.: A 2,326-year tree-ring record of climate variability on the northeastern Qinghai-Tibetan Plateau, Geophysical Research Letters, 30, 10.1029/2003gl017425, 2003.

Zhang, Y., Kong, Z., Yan, S., Yang, Z., and Ni, J.: ''Medieval Warm Period'' on the northern slope of central Tianshan Mountains, Xinjiang, NW China, Geophysical Research Letters, 36, 1-5, 10.1029/2009GL037375, 2009.

Zhao, C., Yu, Z., and Ito, E.: Possible orographic and solar controls of Late Holocene centennialscale moisture oscillations in the northeastern Tibetan Plateau, Geophysical Research Letters, 36, L21705, 10.1029/2009GL040951, 2009.

---

## Author Comment (AC3)

**Author response to Anonymous Referee #3**

**Article ID: cp-2021-137**

**General comments:**

This is an interesting case study dealing with the reconstruction of paleoclimatic change in arid Central Asia (ACA) over the past millennium, based on a new lacustrine archive from Lake Dalongchi (Tien Shan). The main objective of this study is to decipher the respective contributions of internal and external driving mechanisms of hydroclimatic variability in ACA, which may help to better understand the chain of reactions involved and better constrain future climate model simulations. If the topic is of interest, I find that the manuscript (in its present state) has several flaws, which require some further consideration and development (especially in the Results and Discussion) before it could be accepted for publication. Hence I would recommend major revisions, and I would like to see a revised version of the manuscript before final acceptance.

We would like to thank the reviewer very much for the valuable and constructive comments. We believe that the quality of our manuscript will be substantially improved by these comments and suggestions. Accordingly, we have prepared detailed point-by-point responses below (marked in *blue*). Line numbers and revised figure numbers that refer to the changes in the revised manuscript version have been marked in *red*.

**Major comments and concerns:**

1.   Lines 38-40: Some key references are missing in the state-of-the-art of the Introduction.
See for instance the recent paleolimnological contribution of Rousseau et al., 2020 (https://doi.org/10.1016/j.palaeo.2020.109987) depicting the sequence of glacier fluctuations and associated palaeoclimatic changes over the past millennium in the Tien Shan mountains. See also Zhang et al., 2009; doi: 10.1029/2009gl037375.
Furthermore, other recognized contributions focusing on hydroclimatic changes in ACA during the Holocene are curiously eluded in the Introduction, although they are crucial in understanding the mechanisms at work at decadal to centennial and longer timescales, (e.g., amongst others Mathis et al., 2014; Lauterbach et al., 2014; Huang et al., 2014, Schwarz et al., 2017 and more recently Sorrel et al., 2021).Hence I feel that the introduction lacks, in particular, a concise but general overview focusing on humidity changes in ACA during the Holocene, as a brief introduction of the

mechanisms controlling hydroclimate changes in this region. Here, a couple of key references are therefore required in the revised version.

Thank you for the constructive suggestion. we have revised the introduction by adding more records about humidity evolution over the last millennium(Rousseau et al., 2020; Zhang et al., 2003; Zhang et al., 2009; Ma et al., 2008) and an overview of the driving mechanisms during the Holocene (Sorrel et al., 2021; Huang et al., 2014; Schwarz et al., 2017; Mathis et al., 2014; Lauterbach et al., 2014; Chen et al., 2019; Chen et al., 2010; Aichner et al., 2015). (Lines 34-79)

2.    Lines 42-43: Can you be more precise and detail what is involved behind the general statement "internal climate variability"? Very imprecise. This is important as you build most of your Discussion on this issue.

Thanks for the comment. The "internal climate variability" refers to the major atmosphere-sea interaction modes. In this paper it mainly refers to the NAO, AMO, or ENSO. We have added some words to the text. (Lines 62-68, 84-85)

3.    Lines 60-62: As evaporation clearly predominates on precipitation (rain, snow) and riverine inputs in the annual hydrological budget of Lake Dalongchi, are there available information regarding the groundwater contribution on the hydrological balance (which could be very high in such lacustrine systems)?

Thanks for this comment. As Dalongchi Lake is located in a remote region, and its catchment are very small, there is no detailed information regarding the groundwater contribution. The hydrogeology information in the catchment we can provide is only that the main aquifer is hosted by ophiolitic rocks and categorized as a fissured rock aquifer (http://gis.geoscience.cn/website/hg/viewer.htm). This aquifer has multi-scale hydraulic discontinuities and low groundwater potentiality due to the structural heterogeneities of the ophiolitic rocks (Lods et al., 2020; Jeanpert et al., 2019; Boroninaa et al., 2003). We added this information in lines 94-98. As the groundwater is generally stable, we thought that the groundwater has a negligible impact on the hydrological balance of Lake Dalongchi.

4.    Lines 66-67: Do you have more clues about which "shrubs" predominate on the northern slope? Which species precisely?

Yes! During the field work in August 2021, we found that the northern slope is covered by herbs such as Chenopodiaceae, *Artemisia*, Poaceae, and Cyperaceae. We have added it in the manuscript. (Lines 109-110)

5.  Line 66: Correct "southern" and "western".

Thanks. We have revised the manuscript accordingly. (Line 109)

6. Line 67: Correct "northern"

Thanks. We have revised the manuscript accordingly. (Line 109)

7. Results, Chronology, lines 107-113: There is no description provided for core lithology. Are they some hiatus identified in the studied core? I see on Figure 2 that part of the core is laminated, while other intervals look more homogeneous. Hence changes in sedimentation rates should be expected over the past millennium. This is of importance because the authors state (in the Abstract, in the Introduction and in the Conclusions, but never in the main part of the text, why not developed any further in the Results?) that their age model has a very high and constant resolution of ca. 1,8 year (!). Hence some more detail regarding lithological change and sedimentation rates should be provided, and developed, in this chapter.

Thanks for the comment. The descriptions for the core lithology were provided in the 3.1 in the original manuscript. In the revised manuscript, we moved the lithology results to 4.1 (Lines 147-161), and added changes in sedimentation rates in Fig. 2. The dramatic lithological variations and the unstable sediment accumulation rates in Unit B well support the fact that the climate is instability during the LIA indicated by HI. We added this in the discussion section. (Lines 230-232)

8. Line 126: "... as has a steep headwall": More detail should be provided in the study site. By the way, we are not provided with any clue regarding the geological setting of the formations surrounding the lake, in particular in the catchment from which most of the inputs originate. Please provide some more emphasis on this.

Thanks for the comment. We have moved it to the "Study site", and revised to "Both the north and south sides of the lake are surrounded by steep mountains". (Lines 100-101)
Regarding to the geological setting, the bedrock in the catchment of Lake Dalongchi is composed of pillow lava, gabbro and limestone blocks included in a matrix of sheared calcareous turbidites. The ophiolitic mélanges are juxtaposed against Paleozoic sedimentary rocks, gneissic granitoids, and andalusite cordierite-bearing micaschist (Xiao et al., 2013; Ma et al., 2006; Gao et al., 1998). We added these in section "Study site" (Lines 91-94)

9. Line 130: There must be a mistake here: the distance between the catchment and the lakeshore should be lower during high stands (compared to low stands). Correct it.

Regarding this question, maybe we have a problem with expression, easy to cause the

misunderstanding to others. The distance we mean is the distance between the lakeshore and sampling site rather than between the lakeshore and the catchment. Given that the weak inflows of the runoff into Lake Dalongchi which is a shallow and a small lake with an area only of 1.4 km$^2$, the distance from the lakeshore to the sampling site is the key to determining the amount of exogenous detrital materials in the core. The detrital materials are easily deposited in the sampling site on the low lake level conditions with a short distance between the sampling site and lakeshore and vice versa. Please see Lines 182-192.

10. Lines 133-134: This statement is not really convincing as one could expect higher riverine inputs (and thus higher magnetic susceptibility or MS, high silt content and higher C/N ratios) during more humid intervals (rather than during dry intervals). What do the pollen say at the local and regional scale? Are there existing and available palynological data, which would favour one of those two hypotheses?

Thanks for the comment. Please see our response to your comment 9. The detrital materials are easily deposited in the sampling site on the low lake level conditions with a short distance between the sampling site and lakeshore and vice versa. Therefore, during the humid/dry period represented by high/low lake level and enlarged/reduced lake area, exogenous materials containing magnetic minerals, coarse grain components, and terrestrial plants were poorly/easily transported to the sampling site. Pollen analysis is a time-consuming work and is still in progress. Our preliminary palynological data from 18 samples in different depths of the core DLC1819 show that a dry climate characterized by herb pollen (~71 %) dominated by *Artemisia*, Chenopodiaceae, and Poaceae during in the MWP, and a wet climate characterized by the rapid increased tree pollen dominated by the Picea (up to 45%) during in the LIA. Please see *the following Figure 1* and in the revised manuscript. (Lines 204-207). Our preliminary palynological result supports the interpretation of multi-proxies in the manuscript, i.e., higher MS, higher silt content and higher C/N ratios reflect more dry climate and vice versa.

[Figure]

*Figure 1. The preliminary results of pollen analysis in DLC1819 core. The light blue rectangle represents the Little Ice Age.*

11. Lines 139-140: What is the impact of wind activity around the lake? Is it possible to discriminate between aeolian and riverine inputs in the clastic fraction? Any smear slide analysis performed to check the shape/texture of minerogenic grains? By the way, what is the grain mineralogy of "exogenous materials"? Quartz? Oxides? Else? More information should definitely be provided in this section (and thus also ahead in the Study site section regarding the catchment). Besides, XRF data would have been of help to identify grain size variations, possible sources and discuss changes in clastic inputs over time. Any possibility to add such a dataset in a revised manuscript?

Thanks for your nice suggestions. The prevailing wind is from the northwest and the annual average wind speed is only 2.73 m/s. The average number of sandstorm days is less than 5 days/year based on the meteorological records from 1959 to 1998 (Zhou et al., 2002; Zhang et al., 2021a) (Lines 115-117). In fieldwork, we also found that there is almost no aeolian deposition (no loess and sand desert accumulation) in the basin. So, the wind activity has a little influence on Lake Dalongchi.

According to your insightful suggestion, we randomly selected four samples sifted by 30-mesh and 115-mesh sieve after removing organic matter and carbonates, and observed them under the microscope. We found that the grains in the range of 125-500 μm are characterized by poor roundness with an angular outline which is quite different from the aeolian materials (Zhang et al., 2021b), excluding the possibility that the clastic particles are derived from aeolian deposition in Lake Dalongchi. Please see the *following Figure 2,* Fig. S1 in the supplement, and lines 174-181 in the revised

manuscript.

[Figure]

*Figure 2. The photomicrograph comparison of 125-500 µm size-fractions in Lake Dalongchi and the typical eolian sand in Jilantai Salt Lake. (a) The 125-500 µm size-fractions from randomly selected samples of DLC1819 core. (b) The typical eolian sand from randomly selected samples of JLT-2010 core (Zhang et al., 2021b).*

The grain mineralogy could not distinguish between aeolian and riverine inputs, but we also randomly selected 4 samples in different depths of the core for mineral analysis by X-ray diffraction. The results show that the minerals of Lake Dalongchi sediments are mainly composed of quartz, illite, albite, calcite, and clinochlore (*please see the following Figure 3*). Regarding the XRF analysis, we do not have XRF data at present, but we will do XRF core scanning on a longer core (ca. 15 m) that we plan to get from Lake Dalongchi in September 2022. Thank you much for your suggestion and hope we can get good XRF results in the future.

[Figure]

*Figure 3. The XRD diffractograms and mineral compositions of 4 randomly selected samples in DLC1819 core*

12. Figure 4: Please add the impact of wind processes on the two cartoons, i.e., during High stands and low stands.

Thanks for the comment. The wind activity can be ignored as the the wind speed is low and there no aeolian deposition in catchment of Lake Dalongchi, so we do not consider the wind effect on lake deposition and on the climate reconstruction.

13. Lines 143-145: I doubt that the relationship/consistency between the reconstructed HI and the instrumental relative humidity can be used that straight, since the correlation (as well as the R2; R2=0.298) are rather low. This statement should thus be tempered down.

Thanks for this reminder. We have revised our statement in the manuscript as follows: "There is a generally positive correlation (r = 0.298*) between the reconstructed HI and the instrumental relative humidity records over the past 60 years from the nearby Bayanbuluk meteorological station at the 0.05 significant level, verifying the reliability of the humidity reconstruction." (Lines 200-202)

14. Lines 145-149: Here the results should be compared with those recently published by Rousseau et al., 2020 (Palaeo3) (https://doi.org/10.1016/j.palaeo. 2020.109987) relying on a proglacial lake from the Tien Shan mountains in Kyrgyzstan over the past millennium. Such a comparison should also be integrated on Fig. 6.

Thank you very much. We have added it to Fig. 6 and revised the discussion in the part of "The humidity changes over the last millennium". (Lines 214-217, Fig. 6h)

15. Line 161: Replace "during in MWP to LIA" into "between the MWP and the LIA".

Thanks! We have corrected it. (Line 264)

16. Figure 5: This figure (vs age) is finally very similar to Figure 4 (vs depth). Perhaps you could highlight more clearly (with vertical bands) the most important time slices for the Discussion (driest / humid intervals) on Figure 5.

Thanks for your comment. We have added the humid interval during the LIA indicated by the light grey bars in Fig. 5.

17. Lines 162-166: The authors shortly state the discrepancies between the HI index developed in this study and other humidity records in ACA, but strangely do not provide any reason for it. Then, how would you explain such discrepancies between the different ACA records? We are in the Discussion; hence this should at least be developed a minima (and I would not expect the respective age models to account for the differences observed). Very important.

Thanks for the comment. On the multi-millennial, the hydroclimate changes revealed by our reconstruction are generally in agreement with the hydroclimatic patterns revealed by recent numerous studies in ACA (Chen et al., 2006; Song et al., 2015; Lan et al., 2018; Lan et al., 2019; Zhao et al., 2009; He et al., 2013; Ma and Edmunds, 2006; Gates et al., 2008; Rousseau et al., 2020). (Lines 207-219)
The previous study shows that the higher anomalous climatic instability during the LIA compared to the MWP, suggesting the moisture instability prefers to occur within the conditions of an overall cold climate (Chen et al., 2019b). However, it is not clear how the specific unstable wet and dry climate fluctuated during the LIA, due to the relative low-resolution records in ACA (Chen et al., 2006; Zhao et al., 2009; Lan et al., 2019). The HI reconstruction of Lake Dalongchi provides new evidence for the unstable hydroclimatic variability during the LIA (Fig. 6b). Our high-resolution reconstruction clearly documented several obvious and dramatic secondary humidity fluctuations within the LIA, which are not clearly captured in other current records from ACA (Chen et al., 2006; Ma and Edmunds, 2006; Gates et al., 2008; He et al., 2013) (Fig. 6). The climatic instability during the LIA can also be reflected by the dramatic lithological variations and the unstable sediment accumulation rates in Unit B (Fig. 2) (Lines 220-232)

18. Figure 6: Where is the Badain Jaran locality? Sugan Lake? Lake Gahai? Which country in ACA? This should occur on Figure 1, or on a separate panel in Figure 6.

Thanks! The extent of the ACA (arid Central Asia) is essentially equivalent to the mid-latitude inland region on the western side of the modern Asia summer monsoon. We

have annotated the records mentioned in the text in Figure 1 and added the explanation of the ACA. (Lines 700-706, Fig. 1)

19. Line 176: Here it is stated that periodicities of coherence occur from 88 to 146 years, although 88 to 157 years are quoted line 168. Please clarify it.

Thanks! We have revised the "88 to 157 years" to the "~88-146 years". (Line 233, 275, 278)

20. Lines 188-194: Here again we are in the Discussion, not in the Introduction. This paragraph is in fact almost devoid of any information, as we are only provided with very general statements mentioning that a solar forcing was also involved in other regional records, but without providing any clues regarding the chain of reactions and/or the mechanisms at work behind (at least an attempt could have been done). Such a relationship between solar activity and lake proxies has been long reported in the literature over the past 30 years, but we do not learn much more here. This part of the manuscript would deserve a more in-deep discussion and some more development.

We appreciate the constructive suggestion. The possible physical mechanisms to the hydroclimate changes in ACA of the Solar force are very important. The discussion has been substantially reorganized in the following aspects.

(1) We used the Ensemble empirical mode decomposition (EEMD), a new noise-assisted data analysis (NADA) method (Wu and Huang, 2009; Huang et al., 1998), to extract the century-signal from the original HI series. The results show that there is a firm negative relationship between the century-component of the HI and solar irradiance, which verifies the critical role of the Gleissberg solar cycle in controlling the effective humidity at the century-scale during the last millennium in ACA. (Lines 280-287, Fig. 7f)

(2) We added more discussion regarding several records from ACA that documented the solar fingerprint (Zhao et al., 2009; Yin et al., 2016; Ling et al., 2018; Song et al., 2015). However, rare records in ACA documented the good relationship between the effective humidity changes and the fluctuations of the Gleissberg cycle, even though such records exhibited periodicities of 93 years and 70 to 100 years through the Spectral and wavelet analysis. (Lines 292-298)

(3) Next, we focus on the confusion how solar activity significantly affected hydroclimate fluctuations in ACA over the last millennium. The indirect mechanism is that the solar variability indirectly affects the hydroclimate changes through modulating the NAO state, suggesting solar regulation for hydroclimate might be amplified on a regional scale through atmospheric circulation (Ineson et al., 2011; Shindell et al., 2001; Gray, 2003; Brahim et al., 2018; Kodera, 2002; Yukimoto et al., 2017). However, the direct mechanism of solar forcing through modulating the evaporation seems to have a greater effect on ACA with scare precipitation and intense evaporation. Thus, we preliminarily proposed that solar activity has a direct influence on the effective humidity through controlling regional evaporation in ACA. We further performed the transient experiment forced only by the TSI for

the last millennium using the Max Planck Institute Earth System Model (MPI-ESM) (Jungclaus et al., 2014), to investigate the potential feedback processes between the solar variability and effective humidity at century timescales in ACA. The result also indicates that solar irradiance has an important contribution to the humidity changes of ACA by regulating the temperature and evaporation. (Lines 301-332, Fig. S4 in supplement)

21.  Lines 196-219: Same comment here regarding the link between the HI and the ENSO. Even if wavelet analyses suggest a negative relationship between the HI and ENSO, this is however tricky to see any kind of correlation (or anticorrelation) between the two datasets. At least, kind of a correlation could be observed after 1800 AD, but interestingly not before. How would you account for that?

We thank the valuable suggestion. There is a negative phase relationship between the HI and ENSO variance at multidecadal timescales. However, this relationship can only be revealed by the WTC results, rather than the two datasets, because the original HI contains a variety of signals at different timescales. To solve such a problem, we further performed the ensemble empirical mode decomposition (EEMD), a new noise-assisted data analysis method (Huang et al., 1998; Wu and Huang, 2009), to extract the multidecadal signals of the HI. Interestingly, the extracted multidecadal component of the HI exhibits better inverse relationship with the ENSO variance almost throughout the entire time series (please see the *following Figure 4*), which is in line with the WTC results. More discussions were added in the revised manuscript. (Lines 358-377, Fig. 7g and Fig. S3b in the Supplement)

[Figure]

*Figure 4. The comparison between the multidecadal component of HI extracted by EEMD and ENSO variance (Li et al., 2011).*

22.  But, basically, I am puzzled again by the fact that we do not learn much more at the end of the manuscript that what has been widely elsewhere in the literature, especially regarding the driving mechanisms of hydroclimatic variability in ACA during the late Holocene. Hence I would recommend to revise the Discussion by bringing a far stronger case on proxy correlation between the different regional records presented in Figure 6, as when tackling the possible mechanisms at work controlling climate variability over the timespan studied.

Thanks for the comment. Regarding Fig. 6, the comparison among the records can only

be on the multi-centennial timescale scales (i.e., WMP, LIA), and this has been discussed by a large number of previous studies. However, there are rarely multidecadal-resolution sediment records in ACA due to the commonly low sedimentation rate and old carbon effect. Therefore, our reconstruction is difficult to compare with other records from ACA at multidecadal to century scales during the last millennium. The purpose of the CWT, WTC and EEMD analysis based on our reconstruction is to reveal the potential mechanisms at different timescales: the dry climate during the MWP and wet climate during the LIA at multi-centennial timescales are mainly attributed to the influence of the NAO and AMO, and the humidity oscillation is directly modulated by the Gleissberg solar cycle at the century-scale and by the quasi-regular period of ENSO at the multidecadal Scale. We also expected more high resolution records to confirm or debate.

Overall, based on your and other two reviews suggestions and comments, we have made substantial revisions by adding EEMD data analysis, sensitivity experiment, and more meaningful previous researches. Four supplementary figures were added in the supplement of the manuscript. Thank you very much for your helpful comments again.

**Reference**

Aichner, B., Feakins, S. J., Lee, J. E., Herzschuh, U., and Liu, X.: High-resolution leaf wax carbon and hydrogen isotopic record of the late Holocene paleoclimate in arid Central Asia, Climate of the Past, 11, 619-633, 10.5194/cp-11-619-2015, 2015.

Boroninaa, A., Renarda, P., Balderera, W., and Christodoulides, A.: Groundwater resources in the Kouris catchment (Cyprus): data analysis and numerical modelling, Journal of Hydrology, 271, 130-149, 10.1016/S0022-1694(02)00322-0, 2003.

Brahim, Y. A., Wassenburg, J. A., Cruz, F. W., Sifedine, A., Scholz, D., Boumchou, L., Dassie, E. P., Jochum, K. P., Edwards, R. L., and Cheng, H.: Multi-decadal to centennial hydroclimate variability and linkage to solar forcing in the Western Mediteraanean during the last 1000 years., Scientific Reports, 8, 1-8, 2018.

Chen, F., Huang, X., Zhang, J., Holmes, J. A., and Chen, J.: Humid Little Ice Age in arid central Asia documented by Bosten Lake, Xinjiang, China, Science in China Series D: Earth Sciences, 49, 1280-1290, 10.1007/s11430-006-2027-4, 2006.

Chen, F., Chen, J., Holmes, J., Boomer, I., Austin, P., Gates, J. B., Wang, N., Brooks, S. J., and Zhang, J.: Moisture changes over the last millennium in arid central Asia: a review, synthesis and comparison with monsoon region, Quaternary Science Reviews, 29, 1055-1068, 10.1016/j.quascirev.2010.01.005, 2010.

Chen, F., Chen, J., Huang, W., Chen, S., Huang, X., Jin, L., Jia, J., Zhang, X., An, C., Zhang, J., Zhao, Y., Yu, Z., Zhang, R., Liu, J., Zhou, A., and Feng, S.: Westerlies Asia and

monsoonal Asia: Spatiotemporal differences in climate change and possible mechanisms on decadal to sub-orbital timescales, Earth Science Reviews, 192, 337-354, 10.1016/j.earscirev.2019.03.005, 2019.

Gao, J., Li, M., Xiao, X., Tang, Y., and He, G.: Paleozoic tectonic evolution of the Tianshan Orogen, northwestern China., Tectonophysics, 287, 213-231, 10.1016/S0040-1951(98)80070-X, 1998.

Gates, J. B., Edmunds, W. M., Ma, J., and Sheppard, P. R.: A 700-year history of groundwater recharge in the drylands of NW China, The Holocene, 18, 1045-1054, 10.1177/0959683608095575, 2008.

Gray, L. J.: The influence of the equatorial upper stratosphere on stratospheric sudden warmings, Geophysical Research Letters, 30, 10.1029/2002gl016430, 2003.

He, Y., Zhao, C., Wang, Z., Wang, H., Song, M., Liu, W., and Liu, Z.: Late Holocene coupled moisture and temperature changes on the northern Tibetan Plateau, Quaternary Science Reviews, 80, 47-57, 10.1016/j.quascirev.2013.08.017, 2013.

Huang, N. E., Shen, Z., Long, S. R., Wu, M. C., Shih, H. H., Zheng, Q., Yen, N.-C., Tong, C. C., and Liu, H. H.: The empirical mode decomposition and the Hilbert spectrum for nonlinear and non-stationary time series analysis, Proceedings of the Royal Society of London, 454, 903-995, 10.1098/rspa.1998.0193, 1998.

Huang, X., Oberhänsli, H., von Suchodoletz, H., Prasad, S., Sorrel, P., Plessen, B., Mathis, M., and Usubaliev, R.: Hydrological changes in western Central Asia (Kyrgyzstan) during the Holocene as inferred from a palaeolimnological study in lake Son Kul, Quaternary Science Reviews, 103, 134-152, 10.1016/j.quascirev.2014.09.012, 2014.

Ineson, S., Scaife, A. A., Knight, J. R., Manners, J. C., Dunstone, N. J., Gray, L. J., and Haigh, J. D.: Solar forcing of winter climate variability in the Northern Hemisphere, NATURE GEOSCIENCE, 4, 753-757, 10.1038/NGEO1282, 2011.

Jeanpert, J., Iseppi, M., Adler, P. M., Genthon, P., Sevin, B., Thovert, J. F., Dewandel, B., and Join, J. L.: Fracture controlled permeability of ultramafic basement aquifers. Inferences from the Koniambo massif, New Caledonia, Engineering Geology, 256, 67-83, 10.1016/j.enggeo.2019.05.006, 2019.

Jungclaus, J. H., Lohmann, K., and Zanchettin, D.: Enhanced 20th-century heat transfer to the Arctic simulated in the context of climate variations over the last millennium, Climate of the Past, 10, 2201-2213, 10.5194/cp-10-2201-2014, 2014.

Kodera, K.: Solar cycle modulation of the North Atlantic Oscillation: Implication in the spatial structure of the NAO, Geophysical Research Letters, 29, 59-51-59-54, 10.1029/2001gl014557, 2002.

Lan, J., Xu, H., Sheng, E., Yu, K., Wu, H., Zhou, K., Yan, D., Ye, Y., and Wang, T.: Climate changes reconstructed from a glacial lake in High Central Asia over the past two

millennia, Quaternary International, 487, 43-53, 10.1016/j.quaint.2017.10.035, 2018.

Lan, J., Xu, H., Yu, K., Sheng, E., Zhou, K., Wang, T., Ye, Y., Yan, D., Wu, H., Cheng, P., Abuliezi, W., and Tan, l.: Late Holocene hydroclimatic variations and possible forcing    mechanisms over the eastern Central Asia, Science China, 62, 1288-1301, 10.1007/s11430-018-9240-x, 2019.

Lauterbach, S., Witt, R., Plessen, B., Dulski, P., Prasad, S., Mingram, J., Gleixner, G., Hettler-Riedel, S., Stebich, M., Schnetger, B., Schwalb, A., and Schwarz, A.: Climatic imprint of the mid-latitude Westerlies in the Central Tian Shan of Kyrgyzstan and teleconnections to North Atlantic climate variability during the last 6000 years, The Holocene, 24, 970-984, 10.1177/0959683614534741, 2014.

Li, J., Xie, S.-P., Cook, E. R., Huang, G., D'Arrigo, R., Liu, F., Ma, J., and Zheng, X.-T.: Interdecadal modulation of El Niño amplitude during the past millennium, Nature Climate Change, 1, 114-118, 10.1038/NCLIMATE1086, 2011.

Ling, Y., Dai, X., Zheng, M., Sun, Q., Chu, G., Wang, H., Xie, M., and Shan, Y.: High-resolution geochemical record for the last 1100 yr from Lake Toson, northeastern Tibetan Plateau, and its climatic implications, Quaternary International, 487, 61-70, 10.1016/j.quaint.2017.03.067, 2018.

Lods, G., Roubinet, D., Matter, J. M., Leprovost, R., and Gouze, P.: Groundwater flow characterization of an ophiolitic hard-rock aquifer from cross-borehole multi-level hydraulic experiments, Journal of Hydrology, 589, 10.1016/j.jhydrol.2020.125152, 2020.

Ma, C., Wang, F., Cao, Q., Xia, X., Li, S., and Li, X.: Climate and environment reconstruction during the Medieval Warm Period in Lop Nur of Xinjiang, China, Science Bulletin, 53, 3016-3027, 10.1007/s11434-008-0366-6, 2008.

Ma, J. and Edmunds, W. M.: Groundwater and lake evolution in the Badain Jaran Desert ecosystem, Inner Mongolia, Hydrogeology Journal, 14, 1231-1243, 10.1007/s10040-006-0045-0, 2006.

Ma, Z., Xia, L., Xu, X., Xia, Z., Li, X., and Wang, L.: Geochemical characteristics of basalts: evidence for the tectonic setting and geological significance of Kulehu ophiolite, South Tianshan Mountains, Acta Petrologica et Mineralogica, 25, 387-400, 10.1016/S1872-2040(06)60043-1, 2006. (in Chinese with English Abstract)

Mathis, M., Sorrel, P., Klotz, S., Huang, X., and Oberhänsli, H.: Regional vegetation patterns at lake Son Kul reveal Holocene climatic variability in central Tien Shan (Kyrgyzstan, Central Asia), Quaternary Science Reviews, 89, 169-185, 10.1016/j.quascirev.2014.01.023, 2014.

Rousseau, M., Demory, F., Miramont, C., Brisset, E., Guiter, F., Sabatier, P., and Sorrel, P.: Palaeoenvironmental change and glacier fluctuations in the high Tian Shan

Mountains during the last millennium based on sediments from Lake Ala Kol, Kyrgyzstan, Palaeogeography, Palaeoclimatology, Palaeoecology, 558, 10.1016/j.palaeo.2020.109987, 2020.

Schwarz, A., Turner, F., Lauterbach, S., Plessen, B., Krahn, K. J., Glodniok, S., Mischke, S., Stebich, M., Witt, R., Mingram, J., and Schwalb, A.: Mid- to late Holocene climate-driven regime shifts inferred from diatom, ostracod and stable isotope records from Lake Son Kol (Central Tian Shan, Kyrgyzstan), Quaternary Science Reviews, 177, 340-356, 10.1016/j.quascirev.2017.10.009, 2017.

Shindell, D. T., Schmidt, G. A., Mann, M. E., Rind, D., and Waple, A.: Solar Forcing of Regional Climate Change During the Maunder Minimum, Science, 294, 2149-2152, 10.1126/science.1064363, 2001.

Song, M., Zhou, A., Zhang, X., Zhao, C., He, Y., Yang, W., Liu, W., Li, S., and Liu, Z.: Solar imprints on Asian inland moisture fluctuations over the last millennium, The Holocene, 5, 1935-1943, 10.1177/0959683615596839, 2015.

Sorrel, P., Jacq, K., Van Exem, A., Escarguel, G., Dietre, B., Debret, M., McGowan, S., Ducept, J., Gauthier, E., and Oberhänsli, H.: Evidence for centennial-scale Mid-Holocene episodes of hypolimnetic anoxia in a high-altitude lake system from central Tian Shan (Kyrgyzstan), Quaternary Science Reviews, 252, 10.1016/j.quascirev.2020.106748, 2021.

Wu, Z. and Huang, N. E.: Ensemble Empirical Mode Decomposition: A Noise-Assisted Data Analysis Method, Advances in Adaptive Data Analysis, 1, 1-41, 2009.

Xiao, W., Windley, B. F., Allen, M. B., and Han, C.: Paleozoic multiple accretionary and collisional tectonics of the Chinese Tianshan orogenic collage, Gondwana Research, 23, 1316-1341, 10.1016/j.gr.2012.01.012, 2013.

Yin, Z.-Y., Zhu, H., Huang, L., and Shao, X.: Reconstruction of biological drought conditions during the past 2847 years in an alpine environment of the northeastern Tibetan Plateau, China, and possible linkages to solar forcing, Global and Planetary Change, 143, 214-227, 10.1016/j.gloplacha.2016.04.010, 2016.

Yukimoto, S., Kodera, K., and Thiéblemont, R.: Delayed North Atlantic Response to Solar Forcing of the Stratospheric Polar Vortex, Sola, 13, 53-58, 10.2151/sola.2017-010, 2017.

Zhang, B., Liu, X., and Li, J.: The aeolian component inferred from lake sediments in China, Aeolian Research, 50, 10.1016/j.aeolia.2021.100700, 2021a.

Zhang, M., Liu, X., Yu, Z., and Wang, Y.: Paleolake evolution in response to climate change since middle MIS 3 inferred from Jilantai Salt Lake in the marginal regions of the ASM domain, Quaternary International, 10.1016/j.quaint.2021.06.017, 2021b.

Zhang, Q.-B., Cheng, G., Yao, T., Kang, X., and Huang, J.: A 2,326-year tree-ring record

of climate variability on the northeastern Qinghai-Tibetan Plateau, Geophysical Research Letters, 30, 10.1029/2003gl017425, 2003.

Zhang, Y., Kong, Z., Yan, S., Yang, Z., and Ni, J.: ''Medieval Warm Period'' on the northern slope of central Tianshan Mountains, Xinjiang, NW China, Geophysical Research Letters, 36, 1-5, 10.1029/2009GL037375, 2009.

Zhao, C., Yu, Z., and Ito, E.: Possible orographic and solar controls of Late Holocene centennialscale moisture oscillations in the northeastern Tibetan Plateau, Geophysical Research Letters, 36, L21705, 10.1029/2009GL040951, 2009.

Zhou, J., Wang, X., and Niu, R.: Climate characteristics of sandstorm in China in recent 47 years., Journal of Applied Meteorological Science, 13, 195-200, 2002. (in Chinese with English Abstract)

---

## Author Comment (AC5)

[Figure]

Figure 1. The comparison between the multidecadal component of HI extracted by EEMD and ENSO variance (Li et al., 2011).

**Reference**

Li, J., Xie, S.-P., Cook, E. R., Huang, G., D'Arrigo, R., Liu, F., Ma, J., and Zheng, X.-T.: Interdecadal modulation of El Niño amplitude during the past millennium, Nature Climate Change, 1, 114-118, 10.1038/NCLIMATE1086, 2011.